# Effect of Immersive Virtual Reality-Based Bilateral Arm Training in Patients with Chronic Stroke

**DOI:** 10.3390/brainsci11081032

**Published:** 2021-08-03

**Authors:** Yo-Han Song, Hyun-Min Lee

**Affiliations:** 1Department of Physical Therapy, Seoyeong University, Gwangju 61268, Korea; songyh727@seoyeong.ac.kr; 2Department of Physical Therapy, Honam University, Gwangju 62399, Korea

**Keywords:** bilateral arm training, chronic stroke, immersive virtual reality, manual function test, upper limb sensory function test

## Abstract

Virtual reality (VR)-based therapies are widely used in stroke rehabilitation. Although various studies have used VR techniques for bilateral upper limb training, most have been only semi-immersive and have only been performed in an artificial environment. This study developed VR content and protocols based on activities of daily living to provide immersive VR-based bilateral arm training (VRBAT) for upper limb rehabilitation in stroke patients. Twelve patients with chronic stroke were randomized to a VRBAT group or a normal bilateral arm training (NBAT) group and attended 30-min training sessions five times a week for four weeks. At the end of the training, there was a significant difference in upper limb function in both groups (*p* < 0.05) and in the upper limb function sensory test for proprioception in the NBAT group (*p* < 0.05). There was no significant between-group difference in upper limb muscle activity after training. The relative alpha and beta power values for electroencephalographic measurements were significantly improved in both groups. These findings indicate that both VRBAT and NBAT are effective interventions for improving upper limb function and electroencephalographic activity in patients with chronic stroke.

## 1. Introduction

Stroke has a wide range of sequelae, including sensory abnormalities, impaired visual perception, cognition, and speech, and difficulty in movement or activity due to loss of motor function [1]. Among the various interventions for the recovery of upper limb function in stroke patients, bilateral arm training is particularly effective [2]. Many studies have investigated the therapeutic effect of bilateral upper limb movement in hemiplegic patients [3,4]. However, most have used exercise programs consisting of simple repetitive tasks or upper limb movements that would be unlikely to have a high voluntary uptake rate by patients.

Treatments based on the Bobath concept, proprioceptive neuromuscular facilitation, task-oriented training, constraint-induced movement therapy, and robotic therapy are currently being used as interventions for upper limb function in stroke patients, and research on these treatments is ongoing. However, these treatments involve monotonous and uninteresting tasks and do not encourage patient participation. To address these problems, the treatment may be presented as a task-solving exercise, during which the patient’s body must move, or as a goal-oriented task to maintain the patient’s interest [5,6].

Virtual reality (VR)-based treatment is a positive intervention that can motivate and inspire patients and be a source of enjoyment [7,8]. Although various studies have used VR techniques for bilateral upper limb training [2], most have been only semi-immersive and performed in a restricted environment, for example, playing a VR game on a flat-screen monitor using a controller and a balance board where the visual component is similar to that of a game that might be played in everyday life but without the immersion that exists in a normal environment. To compensate for this, a head-mounted display has been developed to provide immersive VR. These devices can be used in conjunction with a headset and a controller, and when wearing the headset, a VR screen can be viewed, allowing a 360-degree view of the environment. Such an immersion in VR provides indoor experiences that stroke patients could not otherwise experience directly. However, there has been limited research in rehabilitation related to immersive VR [9].

With the recent development of brain science and the development of brain mapping techniques, functional magnetic resonance imaging (fMRI), positron emission tomography (PET), and electroencephalograms (EEG) are being actively studied [10,11,12,13]. Among them, the EEG is a non-invasive method that is economically inexpensive compared to functional imaging research methods of the brain and can directly observe functional changes in the cerebral cortex in a relatively short time and provide valuable information [14]. The EEG is a flow of electrical activity that occurs when signals are transmitted from the nervous system and are used as data to measure brain activity. New technologies such as brain-computer interfaces provide new strategies to improve neuroplasticity in stroke patients [15]. In addition, the EEG is used to evaluate reduced motion performance due to stroke as it occurs variously according to activity situations such as resting state, motion imagination, and motion performance in the sensorimotor area of the cerebral cortex [16,17].

Bilateral upper extremity training activates the motor evoked potential of the affected cerebral hemisphere [18]. When both hands perform the same motion, using the regular activity of the unaffected cerebral hemisphere as a template, the activity of the affected cerebral hemisphere increases, resulting in an improved movement of the affected hand [19]. In addition, it was reported that brain-computer interface robot training had a significant correlation in the functional connectivity change of motor domain (Brodmann area 6) in chronic stroke patients [20]. Although previous studies have been presented to find the neurological mechanism to support the effectiveness of exercise using brain activity equipment, there is insufficient research on the brain activity of stroke patients for virtual reality and bilateral upper extremity training. Therefore, this study intends to investigate brain activity changes using an EEG to clarify the neurological mechanism of movement of both hands.

The existing VR-related research has focused on functional aspects rather than the effects of the intervention on upper limb function or the findings on the EEG [21]. The aim of this study was to determine the effect of an intervention tool that combines an immersive VR system and bilateral upper extremity training on upper limb function and EEG measurements in stroke patients with chronic hemiplegia.

## 2. Materials and Methods

### 2.1. Subjects

The study included 12 stroke patients who were hospitalized at the C Rehabilitation Hospital or G Rehabilitation Hospital in Gwangju City. The study inclusion criteria were a diagnosis of stroke with hemiplegia lasting for at least 6 months, and a score of 24 or higher on the Mini Mental State Examination-Korean (MMSE-K). The exclusion criteria included visual and sensory problems, symptoms of psychological instability, a history of craniotomy for brain surgery, and concomitant musculoskeletal disease with upper extremity involvement.

### 2.2. Study Design

The study protocol was approved by the institutional review board of Honam University (approval number: 1041223-201812-HR-25). Before the collection of any data from the medical records, informed consent was obtained from each subject after they had been provided with a detailed explanation of the objectives of the study and the procedures to be used. The patients were randomly allocated to a VR-based bilateral arm training (VRBAT) group or to a normal bilateral arm training (NBAT) group. Randomization was achieved using the opaque sealed envelope method. All subjects underwent pre-intervention assessments followed by 4 weeks of intervention. The flow of patients through the study is shown in Figure 1.

### 2.3. Measurements

Upper extremity function was assessed by electromyography (EMG), measurement of the activity in the biceps brachii, triceps brachii, and wrist extensor and flexor muscles in the affected side using the manual function test (MFT) and an upper extremity sensory function test. Spontaneous brain waves were measured and compared with brain wave measurements. All measurements were recorded before and after training by two observers.

#### 2.3.1. Manual Function Test

The upper extremity function test is divided into a total of eight items in three areas that assess the recovery status of upper extremity function by scoring arm movements and the ability to perform manipulations in the affected side. The score for each item is recorded as 1 if it can be performed or as 0 if it cannot be performed. The maximum score is 32 points. The test-retest interobserver reliability of this evaluation tool is r = 0.95, and the Cronbach’s alpha value is 0.95 [22].

#### 2.3.2. Upper Extremity Sensory Function Test

Sensory function in the affected arm was evaluated by testing for two-point discrimination, proprioception, and stereognosis. For each type of sensation, the subject scored 2 (perfect score) if they could correctly identify the exact position and type of stimulus, 1 if they could identify only one position or type of stimulus, and 0 if they could not identify either.

#### 2.3.3. Two-Point Discrimination Test

The two-point discrimination test was performed using a Dellon 2-Point Disk-Criminator (Sammons Preston, Bolingbrook, IL, USA), which has a point separation of 10 mm for the fingertips, 20 mm for the thumb, 20 mm for the little finger, and 100 mm for the forearm. The tip of the finger on the palm on the paralyzed side was stimulated using this device while the patient’s eyes were closed. The subject’s score was 2 if the two-point discrimination was normal and 0 if it was not.

#### 2.3.4. Proprioception Test

With the patient’s eyes closed, the examiner moved the paralyzed arm into various positions and asked the patient to move the unaffected arm into the same positions. The subject’s score was 2 if the position of the unaffected arm was correct and 0 if it was not.

The Stereognosis test was performed using a Stereognosis Kit (Sammons Preston), which consists of 17 small items that are commonly used in everyday life, such as pencils and coins, and cards containing drawings of each of the 17 items. With the patient’s eyes closed, items are placed on the palm on the paralyzed side, and the patient was required to match these items with those shown on the cards.

#### 2.3.5. Upper Limb Muscle Activity

Surface EMG signals were recorded using a BIOS-S40 system (Biobrain Inc., Daejeon, Korea) with the biological signal discretized at a frequency of 250 Hz and quantized to 24 bits. Four of the eight bipolar channels were used to measure the biceps brachii, triceps brachii, wrist extensor carpi, and wrist flexor carpi in the affected side [23]. An Ag–AgCl surface EMG electrode with a distance of 20 mm between the internal electrodes was used. The EMG signal measured by the electrode was amplified 10,000 times through an amplifier and transmitted with discretization and quantization via a USB stick to the terminal. After confirming that the raw data were automatically displayed using BioScan software (Biobrain Inc.), the measured data at the terminal were filtered by setting the notch filter to 60 Hz and the bandpass filter to 0.5–250 Hz. The root-mean-square value of the EMG signal from each muscle was calculated for each type of movement and used for analysis [24].

#### 2.3.6. Electroencephalographic Data

The data for the EEG were collected from 19 electrodes attached to the head using a BIOS-S24 system (Biobrain Inc., Daejeon, Korea). The subject’s brain waves obtained at a 256 Hz sampling frequency with a 0.5 to 50 Hz pass filter and 12-bit AD conversion were recorded in a computer. The electrodes were attached to F7, F3, Fz, F4, F8, T3, C3, Cz, C4, T4, T5, P3, Pz, P4, T6, O1, and O2 using the international 10–20 electrode placement method. Reference electrodes were attached to both ears.

Each EEG electrode (Biobrain Inc., Daejeon, Korea) was a plate-shaped disk coated with gold. To minimize contact resistance between the skin and the electrode, the subject’s head was wiped with cotton wool soaked in alcohol to remove any foreign substances before attaching the electrode. An electrode paste was applied to the electrode before it was attached. Each attached electrode was then covered with gauze to fix it securely in place. After attaching the electrodes, the subject was seated in a comfortable chair in a quiet soundproof examination room.

Signals were collected for data processing by measuring a total of 30 times each for 3 min for a spontaneous EEG and 15 s for induced brain waves. The first and last 5 s of the EEG were removed to exclude the possibility of the subject’s attention deteriorating or the influx of miscellaneous waves. The EEG signals were converted from analog to digital and analyzed using the BioScan program (Biobrain Inc.). The signal transmitted to the computer was then analyzed by an expert EEG reader.

A brain mapping program (Biobrain Inc., Daejeon, Korea) was used to visualize the relative alpha and beta values for each EEG region to check visually for changes in activity.

### 2.4. Interventions

#### 2.4.1. VRBAT Group

The VRBAT group underwent immersive VR-based bilateral arm training. This rehabilitation system uses VR-based visual perception-oriented cognitive rehabilitation content (Tion; Human IT Solution, Mokpo-si, Korea) developed by Professor Hyun-min Lee in the Department of Physical Therapy at Honam University and Human IT Solution in 2018 (Figure 2).

The VR content was driven using a high-performance Notebook (Alienware 17R4; Dell, Round Rock, TX, USA). The DK2 Oculus Rift and Oculus Rift controller (Oculus, Menlo Park, CA, USA) were used to provide the immersive VR experience. The VR content consisted of rehabilitation tasks to improve upper limb function, the ability to perform activities of daily living, and visual perception. All contents were configured to be performed while sitting to reduce the risk of cybersickness and falls. Furthermore, the frame drop phenomenon was prevented by using lightmap baking technology.

The VR rehabilitation content consisted of a daily life training component (living room, kitchen, veranda), a visual perception and cognitive rehabilitation component (convenience store), an exercise evaluation component (home refrigerator), and 14 visual perception tasks. Mobile cognitive rehabilitation content (mobile convenience stores) and daily life education videos were included. Each component consisted of the everyday movements, visual perception-oriented cognitive tasks, and evaluation tools used by multidisciplinary rehabilitation teams (neurologists, physical therapists, occupational therapists) working in the stroke rehabilitation field.

The VR tasks performed included everyday activities, such as turning on lights, organizing a chest of drawers, organizing a kitchen, watering plants, and purchasing items at a convenience store. The virtual living room, kitchen, veranda, and convenience store were designed to simulate real environments.

Interventions were performed for 30 min a day, five times a week, for 4 weeks, for a total of 20 sessions. All subjects in the VRBAT group also underwent an hour of conventional rehabilitation per day (Figure 3).

#### 2.4.2. NBAT Group

Subjects in the NBAT group performed tasks similar to those performed by the VRBAT group, such as turning on lights, arranging a chest of drawers, and arranging a kitchen as part of bilateral upper extremity training in a real environment. The interventions were performed for 30 min a day, five times a week, for 4 weeks, for a total of 20 sessions. All subjects in this group also received an hour of conventional rehabilitation per day (Figure 4).

#### 2.4.3. Statistical Analysis

The demographic and clinical characteristics of the subjects were summarized as descriptive statistics. Patient age and months since stroke onset were compared between the VRBAT and NBAT groups using the independent-samples *t*-test. The data were examined for normality using the Shapiro–Wilk test. The values measured before and after intervention were not normally distributed; therefore, the data from the different time points were analyzed using the Wilcoxon signed-rank test. The values obtained before and after training were compared between groups using the Mann–Whitney *U* test. All statistical analyses were performed using SPSS for Windows version 15.0 (IBM Corp., Armonk, NY, USA). A *p*-value < 0.05 was considered statistically significant.

## 3. Results

One subject in each study group did not complete the intervention because of early discharge from the hospital, leaving data for 10 subjects (*n* = 5 in each group) available for analysis. Demographic and clinical characteristics were comparable between the two study groups (Table 1).

Table 2 shows the results of the Wilcoxon signed-rank test analysis before and after treatment for the evaluation performed. Although there was a significant difference in MFT in both groups, there was no significant difference in the two-point discrimination test, proprioception test, stereognosis test, or the EMG test.

Table 3 shows the analysis results of the Mann–Whitney *U* test for group difference values of the evaluated results. There was no significant difference between the groups in MFT, two-point discrimination test, stereognosis test, or the EMG analysis, and only the proprioceptive test showed a significant difference.

The brain activity in two frequency bands, α (8–13 Hz) and β (13–30 Hz), were mapped and extracted in the frequency interval between 1 and 64 Hz. The spectroscopic power differences of all the symmetric electrode pairs in the right and left hemispheres were calculated. As a result, there was a significant difference in α power values in the P3, Pz C3, Cz, and C4 regions before and after the intervention in the VRBAT group (Figure 5). The α and β power values of the NBAT group showed significant differences in activity in the Cz and C4 regions (Figure 6 and Figure 7). The VRBAT group β power values showed significant differences between the C3 and P3 areas (Figure 8).

## 4. Discussion

This study investigated the effects of intervention using the Tion immersive VR rehabilitation program on upper limb function and cerebral activity in patients with chronic stroke. This program has the advantage of allowing stroke patients to perform necessary everyday tasks using both upper limbs in various virtual environments. Ten subjects with chronic stroke were subjected to an immersive VRBAT or NBAT to determine the effects of VR-based rehabilitation training on upper extremity function and cortical activity.

In a study of 32 chronic stroke patients, Subramanian and Levin found a significant increase in reach and grip performance and movement speed regardless of whether the patient was allocated to a VR training group or a conventional training group [9]. Moreover, Sampson et al. (2012) reported that upper extremity motor performance improved after stretch training in a VR environment in one patient with acute stroke and in four patients with chronic stroke [25]. VR training is an interesting and enjoyable intervention for stroke patients that allows them to relearn motor functions in an environment similar to reality [26]. In this study, upper limb function increased in the VRBAT group, which is consistent with previous reports. Morris and Van Wijck compared bilateral arm training with unilateral arm training in 106 stroke patients and found no significant difference in upper limb function between the study groups [27]. Another study showed that bilateral arm training was more effective for improving upper limb function than arm training only on the affected side [27]. In the present study, bilateral arm training was performed on both sides in the VRBAT group using tools encountered in everyday life, and the transfer of human learning probably occurs.

Intact sensory function of the hand is essential in terms of task performance. The skin and proprioceptors located in the hand perceive and recognize the characteristics of objects by touch, two-point discrimination, movement, and three-dimensional recognition capabilities. Furthermore, many joints and small and long muscles of the hand can manipulate various types of objects by making various shapes and movements [28,29]. In this study, there was a slight improvement in the upper limb sensory function test in the VRBAT group, but the between-group difference was not statistically significant; however, there was a statistically significant difference in intrinsic sensory function in the NBAT group. Therefore, direct proprioception may directly affect stimulation of the skin and intramuscular sensory receptors. Yekutiel and Guttman measured the two-point discrimination of proprioception and stereognosis in the paralyzed hand; both showed significant improvement [30]. This finding indicates that the ability to touch, manipulate directly, and carry objects is a factor that improves the sensory function of the upper extremities, as in our VRBAT group. VR also requires the use of a controller to direct upper limb movement. Therefore, it is difficult to improve the direct recognition of objects, such as three-dimensional sensation and two-point discrimination by using VR alone. The functional elements of the upper limb and hand require both visual processing and somatosensory information. In particular, visual information that distinguishes the position of an object and the characteristics of the object in space, the touch sensation in the hand, two-point discrimination, and three-dimensional recognition are required to perform the task accurately [31]). In this study, immersive VR was thought to have no significant effect on the upper limb sensory function test because it did not provide visual feedback on upper limb movements.

There was no significant difference in upper limb muscle activity before or after training within each group or between the two groups. In a study by Sampson et al. [25], bilateral upper extremity training in VR for 45 min per week for 6 weeks resulted in improved shoulder and elbow muscle strength in patients with chronic or acute stroke. In a study by Barker et al., bilateral upper extremity training for 60 min three times a week for 4 weeks using a robotic-assisted device significantly improved the reaching distance and triceps strength in 33 patients with chronic or acute stroke [32]. Yang et al. found a significant improvement in bilateral muscle power in a training group using a robotic-assisted device [33]. Moreover, Whitall et al. found that bilateral upper limb training with auditory feedback focused on the elbow joint flexors and wrist extensors improved muscle strength and increased the range of motion of the shoulder and wrist in stroke patients [34]. The findings of these studies are not consistent with our present observations and may reflect the fact that the intervention periods were longer in the previous studies and that training contained a mixture of intervention methods, such as robotic-assisted devices and auditory feedback. Furthermore, the previous studies included training methods in which a load was applied, whereas, in our study, most of the actions involving manipulating and moving virtual objects in VR that required the use of a controller. Finally, given that training in the NBAT group involved moving only light objects, such as dishes, bowls, and boxes, there was no significant difference in muscle activity in either study group in our study.

In this study, the relative alpha and beta power values for spontaneous EEG activity were measured and analyzed before and after the intervention. In the VRBAT group, the relative alpha power values were significantly different before and after intervention in the P3, Pz, Cz, and C4 regions, as were the relative beta power values at C3 and T3. In the NBAT group, the relative beta power values were significantly different before and after intervention in the Cz and C4 regions. A previous study reported that patients with stroke showed activation in the sensorimotor area on functional magnetic resonance images after performing complex tasks with both hands [35]. Moreover, an association between task-oriented training and activation in the cerebral cortex has been demonstrated [36]. Furthermore, a significant increase in beta waves in the cortical region during stretching movement has been observed in stroke patients [37]. In the present study, the relative beta power values were significantly increased in the sensorimotor areas, such as Cz, C4, and P3, in the VRBAT and NBAT groups, possibly because the intervention in both groups required the use of both upper limbs to perform various tasks in different environments. In particular, the significant increase in relative beta power in the NBAT group may have resulted from actions involving direct manipulation and movement when performing complex tasks in a real environment, as in previous studies.

This study has some limitations. Even if the VR implemented in this study replicated a typical environment encountered in everyday life, it may not have reflected the actual situation or upper limb functions performed during activities of daily living because it was evaluated in a treatment room. Moreover, the sample size was small; thus, it is unclear whether our findings are representative of all stroke patients. Finally, because of the lack of EEG studies on upper limb function using immersive VR, we could only use preliminary studies for reference or comparison. Further studies are needed to develop a standardized framework for bilateral upper limb training based on immersive VR that can be tailored to the type of stroke.

These findings indicate that both immersive VR-based bilateral arm training and conventional bilateral arm training are effective interventions for improving upper limb function and increasing brain activity in patients with chronic stroke.

## 5. Conclusions

This study was conducted by randomly assigning stroke patients six months or more after the onset of the stroke into an immersive virtual reality-based bilateral arm training group and normal bilateral arm training group. There were some differences in the therapeutic effect between the VRBAT group and the NBAT group. Although there were some differences in the VRBAT group and the NBAT group, it was found to positively affect the improvement of upper extremity function in both groups. In the VRBAT group, it was found that the upper extremity movements of the subjects were improved in the immersive virtual reality intervention environment that did not provide visual feedback on their upper extremity movements. This study confirmed that bilateral upper extremity training using immersive virtual reality is an effective intervention method.

## Figures and Tables

**Figure 1 brainsci-11-01032-f001:**
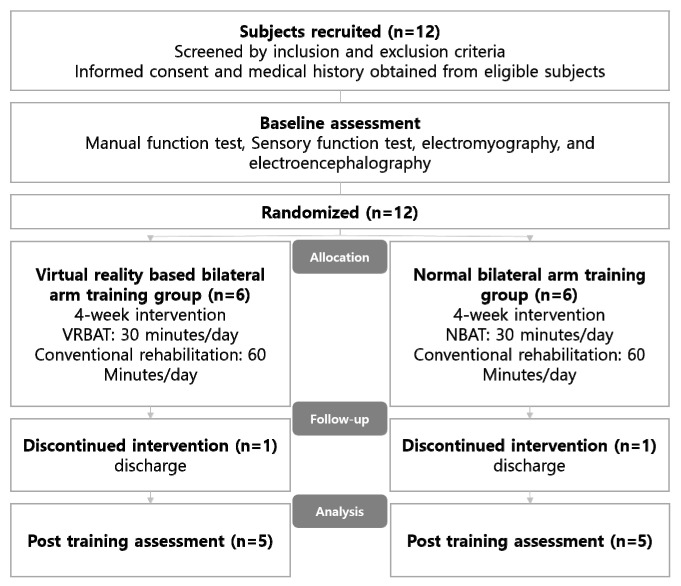
Diagram showing the flow of the study. VRBAT, virtual reality-based bilateral arm training; NBAT, normal bilateral arm training.

**Figure 2 brainsci-11-01032-f002:**
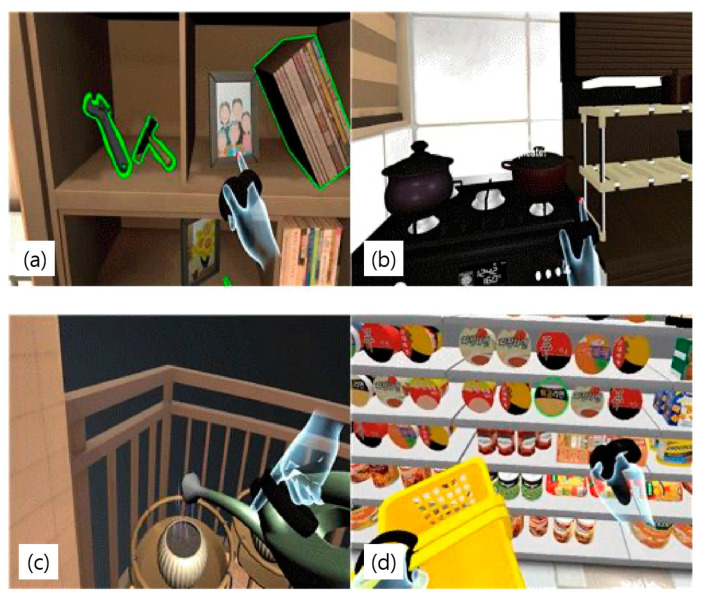
Components of the virtual reality system. (**a**) living room, (**b**) kitchen, (**c**) veranda, (**d**) convenience store.

**Figure 3 brainsci-11-01032-f003:**
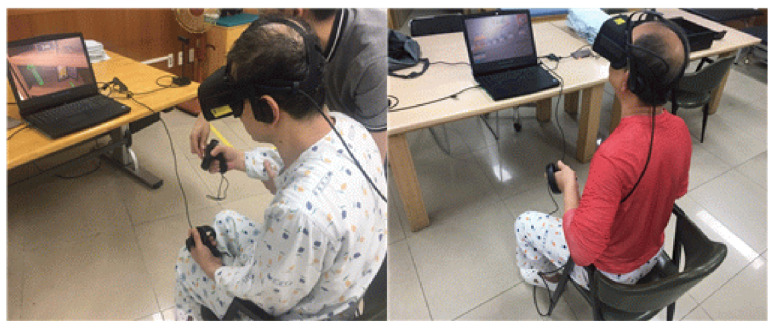
Intervention in the virtual reality-based bilateral arm training group.

**Figure 4 brainsci-11-01032-f004:**
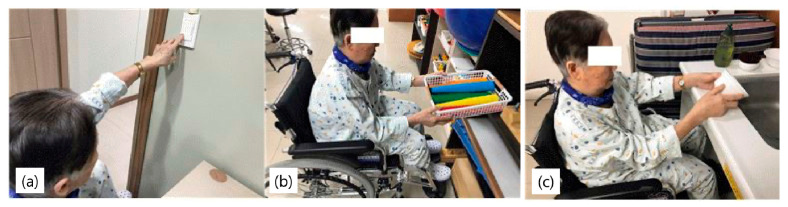
Intervention in the normal bilateral arm training group. (**a**) turning on lights, (**b**) organizing drawers, (**c**) arranging a kitchen.

**Figure 5 brainsci-11-01032-f005:**
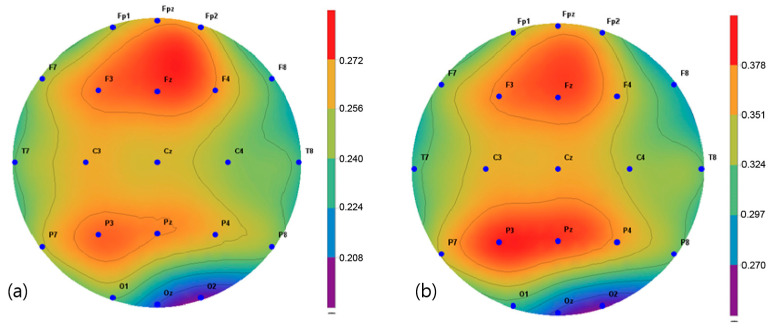
Mapping of the alpha power value (**a**) before and (**b**) after training in the virtual reality-based bilateral arm training group.

**Figure 6 brainsci-11-01032-f006:**
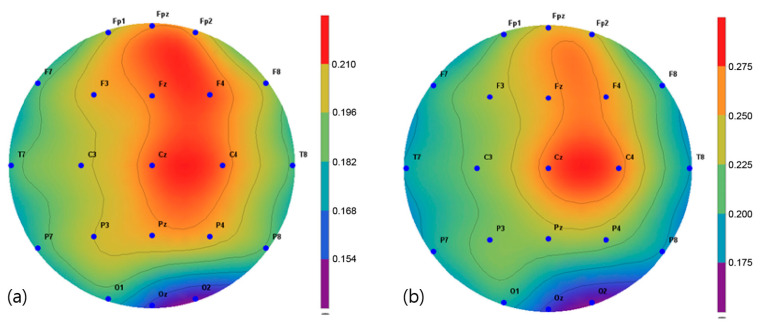
Mapping of the alpha power value (**a**) before and (**b**) after training in the normal bilateral arm training group.

**Figure 7 brainsci-11-01032-f007:**
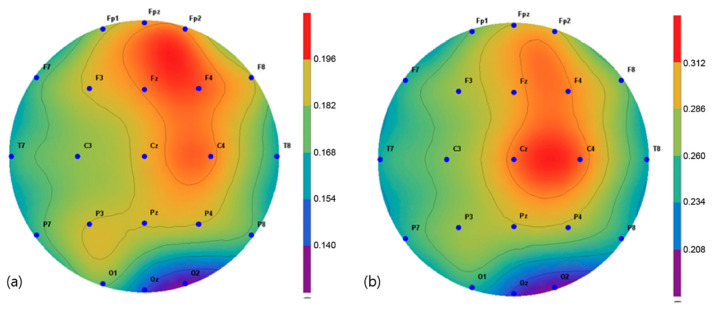
Mapping of the beta power value (**a**) before and (**b**) after training in the normal bilateral arm training group.

**Figure 8 brainsci-11-01032-f008:**
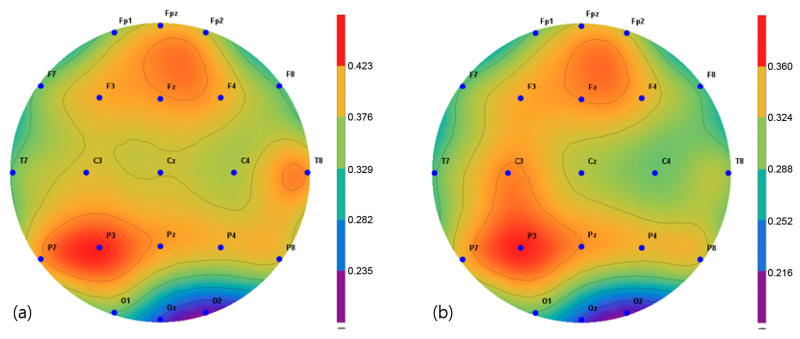
Mapping of the beta power value (**a**) before and (**b**) after training in the virtual reality-based bilateral arm training group.

**Table 1 brainsci-11-01032-t001:** Demographic and clinical characteristics of the study participants.

	VRBAT Group (*n* = 5)	NBAT Group (*n* = 5)
Gender (male/female)	3/2	3/2
Affected side (left/right)	2/3	3/2
Hemorrhagic/Ischemic stroke	2/3	1/4
**Age (years)**	64.20 ± 7.08	60.00 ± 10.88
51–60	2 (20%)	2 (20%)
61–70 y	2 (20%)	2 (20%)
71–80	1 (10%)	1 (10%)
**Post-stroke Duration (months)**	28.40 ± 11.39	25.84 ± 7.34
7–12	0 (0%)	0 (0%)
13–18	1 (10%)	2 (20%)
19–24	3 (30%)	1 (10%)
25–30	1 (10%)	2 (20%)
**MMSE-K (score)**		
24–26	3 (30%)	4 (30%)
26–28	2 (20%)	1 (10%)
28–30	0 (0%)	0 (0%)

MMSE-K, Mini Mental State Examination—Korean version; NBAT, normal bilateral arm training; VRBAT, virtual reality-based bilateral arm training.

**Table 2 brainsci-11-01032-t002:** Wilcoxon Signed-Rank test, pre- and post-training.

	VRBAT Group (*n* = 5)		NBAT Group (*n* =5 )	
	Z	*p*	Z	*p*
MFT	−2.03	0.042 *	−2.07	0.039 *
Two-point discrimination test	−0.67	0.5	−0.4	0.686
Proprioception test	−0.94	0.345	−1.2	0.225
Stereognosis test	−0.94	0.345	−1.2	0.225
Biceps brachii	−1.48	0.225	−0.94	0.686
Triceps brachii	0.94	0.163	−0.94	0.686
Extensor carpi	−0.4	0.893	−0.13	0.686
Flexor carpi	−2.02	0.138	−0.4	0.5

* *p* < 0.05. NBAT, normal bilateral arm training; VRBAT, virtual reality-based bilateral arm training.

**Table 3 brainsci-11-01032-t003:** Intergroup Mann–Whitney *U* Test.

	Paired Differences			
	Mann–Whitney *U*	Wilcoxon W	Z	*p*
MFT	8	23	−0.96	0.07
Two-point discrimination test	10	25	−0.52	0.75
Proprioception test	10	25	−0.52	0.04 *
Stereognosis test	8.5	23.5	−0.83	0.51
Biceps brachii	4	19	−1.77	0.754
Triceps brachii	11	26	−0.31	0.917
Extensor carpi	5	20	−1.56	0.517
Flexor carpi	4	19	−1.77	0.175

* *p* < 0.05.

## Data Availability

Because of the nature of this research, participants of this study did not agree for their data to be shared publicly; thus, supporting data are not available.

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
