# Peer review of "Effect of Immersive Virtual Reality-Based Bilateral Arm Training in Patients with Chronic Stroke"

_brainsci, 2021, doi:10.3390/brainsci11081032_

Round 1
Reviewer 1 Report
Description of statistics analysis should be carefully revised and updated before the next submission.
p.2.1: Subjects description should be carefully updated by pointing out age, antropometric parameters, time after stroke etc....
p.2.3: The present description is very confusing. Specifics should be given to understand the way of data collecting to statistical analyse.
The description of EMG has to be revised and updated. The Authors mentioned "triceps brachii"...which one? What about the reference electrode?
Why the Authors used "the bandpass filter to 0.5250 Hz"???
In which way "The root-mean-square value of the EMG signal from each muscle" was assessed? Data processing approach should be carefully described.
The Authors should carefully explain the protocole used to collect EMG data
p.2.5: Statistical analysis should be carefully rewritten by giving the clear explanation which data had been used in Wilcoxon tests and U Mann-Whitney U test
Table 1 - have to be deeply revised. Some descriptions are very confusing. With respect to the Table 2 - Table 4, analyzed data have to be clear explained
p.3.4: The Authors should carefully explain in which way maps had been created. This information is missed in present description.
Discussion has to be reorganized and Conclusions have to be given.
Supplementary materials are not available online.
Author Response
Dear reviewer,
p.2.1
- Among the general characteristics of the subjects, the elapsed stroke time and age were corrected.
p.2. 3
- Tables and descriptions have been modified for easier understanding.
- The location of EMG electrode attachment for each muscle was additionally described, and references were added.
- The frequency range of the bandpass filter was selected so that the envelope component reflecting muscle activity was included in the EMG time-series signal.
- RMS is an analysis index that reflects the energy of the wave and was used to compare the muscle activity during the corresponding movement.
p.2.4
- The contents related to EEG mapping were corrected and supplemented.
p.2.5
- The data, according to statistical analysis, was corrected and presented in a table.

Reviewer 2 Report
This manuscript evaluates a VR-based intervention for bilateral arm training in people with chronic stroke. While the topic is of interest, I am unsure of the novelty of the results and the manuscript in its current form raises important questions. First, the study sample is very small, in comparison with other studies in the field. This is an important consideration. Furthermore, the manuscript demonstrates some inconsistencies and inaccuracies in writing. The abstract notes that ten patients were randomized, whereas the text describes 12 (with 1 lost to follow-up). The introduction is limited in its description of existing stroke training methods as repetitive and simple – or as monotonous and uninteresting. Furthermore, the extensive literature of VR-based training in chronic stroke is not mentioned. It is unclear why the authors focus on specific EEG outcomes. The selection of specific EMG and EEG measures is unclear from the introduction.
The study included patients with chronic stroke who were >2 years post-stroke on average. However, these patients were hospitalized – it is not clear why? Is it customary for these patients to remain hospitalized for so long? The characteristics of patients lost to follow-up are not analyzed -are they similar to the others? Why were they released early? Table 1 presents unclear and uncomplete data – means (?) are presented without standard deviations (for age and time since stroke). Finally, the interpretation of EEG findings is lacking.
Author Response
Dear reviewer,
- EEG-related contents were added to the introduction.
- The Korean medical system is a system that allows chronic stroke patients after two years to be hospitalized for a long time and receive rehabilitation treatment.
- Two dropout patients in each group were not included in the analysis, and the flow chart was modified according to the relevant items.
- The general characteristics of the subjects in Table 1 were revised again.
- EEG result interpretation part was added.

Round 2
Reviewer 2 Report
I have read the revised version, and while some aspects were improved, I still think that the main issue here, which is that of sample size, is problematic. With N=5 patients in each group I feel that the multiple comparisons performed risk a type 2 error. Furthermore, the authors performed minimal modifications to the text aside from the results section, and I still find the introduction lacking in terms of description of existing studies describing brain activity pre- and post-interventions in people after stroke and highlighting the novelty of the current work. My previous comment re the abstract remains (not ten but 12 participants were randomized). An additional point is that for variables which are not normally distributed, the presentation of means and SDs is misleading and should be avoided.
Author Response
Dear reviewer,
Thank you for reviewing our research.
- We also know that the number of participants in this study is small and increases the type 2 error. So we dealt with the statistics part with utmost care. In a follow-up study currently in progress, the number of subjects has been greatly improved.
- In the introduction, the latest information on stroke patients has been added.
- We corrected the number of subjects in the abstract.
- We deleted the mean and standard deviation from the table.
